# Development and Characteristics of Pancreatic Epsilon Cells

**DOI:** 10.3390/ijms20081867

**Published:** 2019-04-16

**Authors:** Naoaki Sakata, Gumpei Yoshimatsu, Shohta Kodama

**Affiliations:** 1Department of Regenerative Medicine and Transplantation, Faculty of Medicine, Fukuoka University, Fukuoka 814-0180, Japan; gyoshimatsu@fukuoka-u.ac.jp (G.Y.); skodama@fukuoka-u.ac.jp (S.K.); 2Center for Regenerative Medicine, Fukuoka University Hospital, Fukuoka 814-0180, Japan

**Keywords:** ε cell, ghrelin, pancreas, islet, endocrine cell, β cell, Nkx2.2, Pax4, Pax6, insulin

## Abstract

Pancreatic endocrine cells expressing the ghrelin gene and producing the ghrelin hormone were first identified in 2002. These cells, named ε cells, were recognized as the fifth type of endocrine cells. Differentiation of ε cells is induced by various transcription factors, including Nk2 homeobox 2, paired box proteins Pax-4 and Pax6, and the aristaless-related homeobox. Ghrelin is generally considered to be a “hunger hormone” that stimulates the appetite and is produced mainly by the stomach. Although the population of ε cells is small in adults, they play important roles in regulating other endocrine cells, especially β cells, by releasing ghrelin. However, the roles of ghrelin in β cells are complex. Ghrelin contributes to increased blood glucose levels by suppressing insulin release from β cells and is also involved in the growth and proliferation of β cells and the prevention of β cell apoptosis. Despite increasing evidence and clarification of the mechanisms of ε cells over the last 20 years, many questions remain to be answered. In this review, we present the current evidence for the participation of ε cells in differentiation and clarify their characteristics by focusing on the roles of ghrelin.

## 1. Introduction

The pancreas is a unique organ with both exocrine and endocrine functions. Its exocrine function is digestion, which it performs by secreting digestive enzymes from acinar cells into the duodenum via the pancreatic duct. Exocrine tissue (acinar and ductal cells) is the main component of the pancreas, occupying >95% of the organ [1]. The endocrine function of the pancreas involves metabolism management, especially in relation to blood glucose. Pancreatic islets, located in the parenchyma, play a leading role in the production and release of hormones into the bloodstream. Up to the end of the previous century, pancreatic islets were thought to comprise four different cell types—α, β, δ, and pancreatic polypeptide (PP) cells—each producing different hormones. β cells occupy >60% of islet cells [2,3] and are involved in the production and release of insulin, which decreases blood glucose levels. α cells account for approximately 30% of islet cells [2] and produce the hormone glucagon, which acts to increase blood glucose levels. The δ and PP cells are included in the residual 10% of islet cells. δ cells produce somatostatin, a negative regulator of insulin and glucagon secretions under nutritional conditions [4,5,6], while PP cells secrete PP, an inhibitor of glucagon release under low-glucose conditions [7].

A novel type of endocrine cell was identified by Wierup et al. [8] in the early 2000s. These cells, stained with the endocrine marker chromogranin A, did not produce insulin, glucagon, somatostatin, or PP according to immunohistochemistry. However, ghrelin mRNA and protein were detected by in situ hybridization and immunohistochemistry, respectively, in fetal, newborn, and adult human pancreatic islet specimens. Ghrelin-immunoreactive cells were seen at the periphery of the islets, and their locations were independent of other hormone-producing cells. These cells were initially named “ghrelin cells”, but Prado et al. subsequently referred to them as “ε cells” in 2004 [3]. ε cells are now recognized as the fifth type of islet endocrine cell.

## 2. Ghrelin, Regulator of Nutritional Condition

Ghrelin is well known as a “hunger hormone” [1,9]. It was first identified in rat stomachs as an endogenous ligand for the growth hormone secretagogue receptor (GSH-R) in 1999. Ghrelin is a peptide consisting of 28 amino acids with n-octanoylation at serine 3 [10]. It acts as a stimulator for the release of growth hormones from the pituitary via GSH-R, similar to the hormone that stimulates the release of hypothalamic growth hormones [10]. Ghrelin also stimulates the appetite via the hypothalamic arcuate nucleus, which is known to be a regulator of food intake [11]. Expression levels of ghrelin mRNA and protein in the stomach are significantly upregulated by fasting [12]. Neuropeptide Y (NPY) and agouti-related protein (AgRP) are neural peptides that are increased by fasting, and ghrelin induces hyperphagia by activating NPY/AgRP neurons in the arcuate nucleus of the hypothalamus and by accelerating the production of NPY/AgRP proteins [13,14,15]. Ghrelin also suppresses pro-opio-melanocortin (POMC) neurons in the hypothalamic arcuate nucleus and prevents the release of the feeding-deterrent peptide POMC. In contrast, leptin, which is known to work as a mediator of long-term regulation of energy balance by suppressing food intake and inducing weight loss, is released from adipose tissue and exerts the opposite effect to ghrelin on NPY/AgRP and POMC neurons in the hypothalamic arcuate nucleus [16]. Ghrelin and leptin thus compete to regulate energy. In addition to appetite regulation, ghrelin plays various other roles, including stimulating gastric acid release and gastric motility [17], suppressing thermogenesis [18], and having anti-inflammatory and immune-regulatory effects [19].

Ghrelin-producing cells mainly exist in the stomach and especially in the fundus. Circulating ghrelin levels were reduced to 20% after fundectomy in rats [20], and the total ghrelin volume was reduced to 35%–45% after total gastrectomy in humans [21,22]. This indicates that other organs also produce ghrelin and continue to do so after gastrectomy. The small intestine and colon are representative ghrelin-producing organs, although the populations of the relevant cells are smaller than in the stomach [23]. Ghrelin gene expression has also been detected in various organs, including the brain [24], heart [25], lung [26], testis [27], and pancreatic ε cells [8]. Pancreatic ε cells are particularly important sources of ghrelin during human fetal development, but their numbers decrease in adults. This situation is in contrast to the stomach, where ghrelin-producing cells are scarce during pregnancy and increase significantly in adults [8]. Pancreatic ε cells are thought to not only produce ghrelin during pregnancy and to replace the stomach following gastrectomy, but they may also regulate pancreatic hormones produced by other endocrine cells than those that regulate the appetite.

In this review, we summarize the development and characteristics of pancreatic ε cells and consider the roles of ghrelin released from the pancreas.

## 3. ε Cell Development

Ghrelin-producing cells were identified in human pancreatic islets in 2002. Date et al. first demonstrated the presence of ghrelin and detected ghrelin-producing cells in the periphery of human islets, in a similar location to glucagon-producing cells, as in rat islets [28]. Wierup et al. also revealed ghrelin-producing cells in the peripheral rim of the islets at midgestation (18–22 weeks). The population accounted for approximately 10% of islet cells at midgestation, and this decreased to approximately 1% in adults. *Ghrelin* (*GHRL*) mRNA was also detected in the peripheral cells of fetal islets but decreased in adults. In contrast, few ghrelin-producing cells were seen in the stomach at midgestation, but their numbers increased in adults [8]. Andralojc and Sarker’s groups provided further details about the population of ghrelin-producing ε cells. Andralojc et al. observed single ε cells at gestational week 13, which started to aggregate at week 17, surrounded the primitive islets at week 21, and began to decrease at week 23. The population of ε cells accounted for 15% of the islets at week 21, 30% at week 23, 4% at birth, and <1% in adults [29]. On the other hand, Sarker et al. detected ghrelin-producing cells from gestational week 11 [30]. Volante et al. confirmed the presence of ghrelin-producing cells and *GHRL* mRNA in human islets [31]. In mice, several studies revealed ghrelin-expressing cells at embryonic days 8.5–10.5 (E8.5–10.5) [3,32], which is the equivalent of gestational weeks 8–9 in humans [33]. This indicates that ε cells are evident earlier than other islet cell types.

The first step in pancreatic development involves the specification of the primitive endoderm from pluripotent stem cells in blastocysts. This step occurs at E3–5 in mice. Gastrulation to produce the developing ectoderm, mesoderm, and endoderm occurs shortly after specification, and definitive endoderm (DE) cells, which are the origin of pancreas, then form at E6.5–7.5 in mice. The second specification step involves the formation of the posterior gut endoderm, which develops into the midgut and hindgut, from DE cells [34]. Differentiation of the various types of pancreatic cells begins at E8.5 based on the identification of multipotent pancreatic progenitor cells. Expression of the homeodomain transcription factor pancreas/duodenum homeobox protein 1 (PDX-1) is also seen at this time [35]. PDX-1 is an essential factor in the development of acinar, duct, and islet cells. However, although PDX-1 is expressed in exocrine and endocrine progenitors throughout early embryogenesis, it is only expressed in duct progenitors between E9.5 and 12.5 [33,35]. Basic helix–loop–helix transcription factor neurogenin-3 (NGN-3) is another essential factor for the development of endocrine cells, including ε cells [32,33] (Figure 1A). It is first observed in the dorsal pancreatic epithelium at E9, increases from E9.5 to 15.5, and then decreases to a very low level in the neonatal pancreas [36]. Unlike PDX-1, which correlates with the development of exocrine, endocrine, and ductal cells, NGN-3 plays a role in paving the way for differentiation into endocrine progenitors [33]. α cells are first detected in islets at E9.5, followed by β cells within the next 24 h, δ cells at E14.0, and PP cells at E18.0 [37]. Hellar et al. confirmed that NGN-3 was required for the differentiation of endocrine cells. For example, ghrelin/glucagon double-expressing endocrine cells were seen at E18.5 in normal mice, while no ghrelin-producing cells coexpressing insulin, somatostatin, or PP were detected at the same time. The population of ghrelin/glucagon double-producing cells (i.e., α cells) peaked at E10.5 and then decreased during pregnancy. In contrast, the population of cells expressing ghrelin alone (ghrelin^+^glucagon^−^ cells, i.e., ε cells) increased at E15.5 (30%), was maintained during pregnancy, and decreased significantly at birth [32]. Transcription factor V-maf musculoaponeurotic fibrosarcoma oncogene homolog A (MAFA) also contributes to islet development and maturation by binding to the enhancer/promoter region of the insulin gene and driving insulin expression in response to glucose [33]. MAFA is seen at E13.5 but only in insulin-producing cells [38]. Recent studies have revealed that the MAFA level in neonatal islet decreases [39], and development of α and ε cells instead of β cells in pancreatic progenitors occurs in MAFA inhibition under hypothyroidism [40].

The mechanism of why ε cells increase during gestation and decrease before birth is unclear. We consider that the reason lies on the expression patterns of PDX-1, NGN-3, and MAFA. As described previously, PDX-1 and NGN-3 have the role of pancreatic endocrine specification. The peak of the expression level in PDX-1 is at E8–9 (mouse) and week 8 (human) and at E11–12 and weeks 10–11 in NGN-3. MAFA expression begins at this time. In other words, maturation of ε and α cells progresses at this time, while that of β cells just begins. The population of β cells gradually increases following the increase of MAFA expression, which results in the decrease of ε cells.

ε cells are evident earlier than other islet cell types. Several studies have revealed ghrelin-expressing cells at E8.5–10.5 in mice [3,32], indicating that ε cells differentiate earlier than other islet endocrine cells. The key transcription factors involved in ε cell differentiation are Nk2 homeobox (Nkx2.2), paired box protein Pax-4 (Pax4), paired box protein Pax-6 (Pax6), and aristaless-related homeobox (Arx).

Nkx2.2 is a member of the NK2 class of homeodomain transcription factors and is expressed at E8.5–9. Sussel and colleagues examined the contribution of Nkx2.2 to endocrine cell differentiation by immunohistochemistry and revealed that β cells were absent and α and PP cells were reduced in Nkx2.2 knockout mouse embryos [41]. Nkx2.2 plays an important role in β-cell differentiation but also contributes to the differentiation of α and PP cells; however, its role in ε cells is unclear. In 2004, Prado et al. first demonstrated a role for Nkx2.2 in ε cell development using microarray analysis of Nkx2.2^−/−^ mouse pancreas [3] and found no insulin-producing cells and few glucagon-producing cells in the islets of Nkx2.2^−/−^ mice. Moreover, they showed that the population of novel endocrine cells without insulin, glucagon, somatostatin, and PP expression was increased in the mutant islets. Upregulation of both *Ghrl* mRNA and protein were detected in the Nkx2.2 mutant islets (Figure 1). They therefore considered that Nkx2.2 is a key transcription factor involved in directing differentiation toward β or ε cells [41]. They also detected ghrelin-producing cells in the islets of wild-type mice during the prenatal period and showed that this population became very small in adults. During the neonatal stage, approximately one-third of ghrelin-producing cells coexpressed glucagon, while the residual two-thirds only expressed glucagon, but there was no coexpression of ghrelin with insulin, somatostatin, or PP. After this study, the Hill and Sussel group showed that upregulation of ghrelin-producing cells under Nkx2.2 null conditions was not responsible for the loss of insulin-producing cells, and the absence of insulin did not induce the upregulation of ghrelin [42]. Later, they clarified the existence of interaction between Nkx2.2 and NeuroD1, a basic helix–loop–helix transcription factor expressed in the central nervous system, intestine, and pancreas [43,44]. In the study, ε cells and *Ghrl* gene expression were significantly increased in Nkx2.2^−/−^ mouse islets but reduced in Nkx2.2 and NeuroD double knockout islets. In contrast, the absence of Nkx2.2 decreased the expression of glucagon and pancreatic polypeptide, and knockout of NeuroD with partial deletion of Nkx2.2 rescued the numbers of α and PP cells (Figure 1B) [44]. Nkx2.2 receives regulation of NGN-3, and NeuroD1 receives regulation of Nkx2.2. Nkx2.2 decides the development of β or α cells by the activation or prevention of NeuroD under an NGN-3^+^ condition [44]. These results suggest that regulation among Nkx2.2 and NeuroD may contribute to deciding the fate of endocrine cells, including ε cells [44,45]. Wang et al. also examined the effect of Pax4 knockout on the differentiation of ε cells to clarify if the specification from β to ε cells requires Nkx2.2. Pax4 contributes to the differentiation of β cells, and its absence results in a reduction of β cells, similar to Nkx2.2 [46]. Cells expressing ghrelin gene and protein levels were significantly increased in Pax4 knockout mice from E14.5 to birth (Figure 1A). In contrast to Nkx2.2 knockout, a small population of β cells was maintained in Pax4 knockout islets, while the expression of Pax6, which is positively regulated by Nkx2.2, was not affected by Pax4 deletion. They concluded that endocrine progenitors, stimulated by NGN-3, differentiate into β cells in the presence of Nkx2.2 or Pax4 and into ghrelin-producing ε cells in the absence of Nkx2.2 or Pax4. However, the same group later revealed that Nkx2.2 plays a more complex role in the specification to ε cells by demonstrating that Nkx2.2 could bind to and activate the ghrelin promoter [47]. In contrast, Wang and colleagues showed that Pax4 could bind to and repress the ghrelin promoter [48]. The Hill and Sussel group suggested that the upregulation of ghrelin under Nkx2.2 null conditions is not due to the loss of the ghrelin promoter and considered that Nkx2.2 contributes to the activation of ghrelin in mature ε cells [47]. This was supported by Dominguez Gutierrez et al. using single human islet cell RNA sequencing [49]. Regarding Pax6, which is a key factor for generating neural neuronal lineages in the central nervous system and differentiating non-neuronal lineages of the eye [50,51], Swisa and colleague revealed that the absence of Pax6 inhibited production of β cells and induced increase of ε cells using *db/db* mice, a Type 2 DM animal model (Figure 1A) [52]. In summary, the role of Nkx2.2 in the differentiation of ε cells is as follow: (1) deletion of Nkx2.2 contributes to the differentiation of ε cells from endocrine progenitors or other endocrine cells like β cells; (2) the differentiation into ε cells is regulated by interaction with other transcription factors, including NeuroD1; and (3) the maturation of ε cells needs Nkx2.2 expression.

Regarding Arx, Heller and colleagues examined the roles of Arx and Pax4 in ε cell differentiation [32]. Arx is a homeobox-containing gene, notably expressed in nerve and pancreas development. Arx-deficient animals develop hypoglycemia with dehydration as a result of a lack of mature α cells and an increase of β cells [53]. They showed that ghrelin^+^glucagon^+^ α cells were significantly reduced in Arx-mutant pancreas, while ghrelin^+^glucagon^−^ ε cells were still detected (Figure 1). Regarding Pax4, its deletion caused an increase in α cells and losses of β and δ cells [53], while the population of ε cells remained unchanged. They concluded that both Arx and Pax4 upregulated and downregulated the population of α cells but had no influence on ε cells. Importantly, they confirmed that ε cells are regulated by Pax6. ε cells were increased and the α-cell population was decreased in Pax6-mutant islets, while there were fewer ε cells compared with α cells in normal islets (Figure 1). They concluded that the formation of ε cells required NGN-3 expression, similar to the other four types of endocrine cells. Arx and Pax4 contributed to the increase in α-cell but not ε-cell genesis, while the specification of α to ε cells occurred by Pax6 knockout. Mastracci et al. subsequently revealed more detailed roles for Nkx2.2 and Arx in developing ε cells, such as the fact that Arx is not necessary for the specification of ε cells but works on NGN-3^+^ endocrine progenitor cells to regulate ghrelin gene expression under Nkx2.2 null conditions [43].

Finally, ε cells may have a unique role as multipotent progenitors. According to a recent study by Arnes et al., most ghrelin-producing cells change to PP and α cells, while none give rise to β or δ cells. They also showed that ghrelin-producing cells could differentiate into exocrine and ductal cells [54]. This may explain why the population of pancreatic ε cells decreases in adults.

## 4. Role of Ghrelin in Interactions between ε Cells and Other Cells

Ghrelin released from ε cells is recognized as an important factor contributing to the interactions between ε cells and other endocrine cells, with the role of ghrelin-producing ε cells in regulating β cells of particular importance.

Ghrelin is classified as acylated (AG) or unacylated (UAG) according to the presence of absence of acylation of the third serine residue. UAG is also called des-acyl ghrelin [55] (Figure 2A). Acylation is essential for binding ghrelin receptor GHS-R1a and is achieved by enzyme ghrelin O-acyl-transferase [56,57]. GHS-R1a is thus a specific receptor for AG and does not bind to UAG. GHS-R1a has been detected in human and rodent islets and is located in most α cells and some β cells in rats [28,58] and in insulin-producing β cells in humans [59]. Both AG and UAG are derived from cleavage of the N-terminal fragment of prepro-ghrelin. Obestatin (OB) is a novel 23 amino acid *ghrelin* gene-derived peptide (Figure 2A) derived from the C-terminal fragment of the ghrelin precursor [55]. However, the factor that cleaves OB from prepro-ghrelin is still unclear. AG and OB are mainly produced in the stomach but have also been detected in the pancreas [55,60,61,62]. G protein-coupled receptor 39 is considered to be a receptor for OB and is also expressed in human and rodent islets [63,64].

As noted above, regulation of islet function, especially β cells, is an important function of ghrelin released from ε cells. However, the roles of ghrelin in β cells may be complex. Various studies have revealed that ghrelin administration increases blood glucose and decreases plasma insulin in humans and rodents [58,65], and the deletion of ghrelin derived from ε cells significantly enhances glucose-induced insulin release [58,66,67] (Figure 2A). The main contributor to increased blood glucose is AG. The first step in restricting insulin release by AG is activation of GHS-R on β cells, which in turn stimulates the GTP-binding protein Gα_i2_ and suppresses the cAMP signaling pathway in β cells. The decrease of cAMP then induces activation of delayed outward K^+^ (Kv) channels, which inhibits Ca^2+^ channels on β cells and suppresses the influx of Ca^2+^ and insulin release. Ca^2+^ channels are controlled by ATP-sensitive K^+^ (K_ATP_) channels, which are in turn regulated by glucose stimulation [68]. Blockade of ghrelin function and administration of GHS-R antagonists can thus improve the insulin-releasing function. The other mechanism of blood glucose increase by AG involves the stimulation of PP and δ cells, leading to the suppression of insulin release [69,70]. Indeed, it becomes clear that the decrease of insulin release by ghrelin stimulation goes through GHS-R on δ cells rather than β cells (Figure 2B). This phenomenon was first clarified by DiGruccio’s group, which revealed that AG activated GHS-R on δ cells, increased intracellular calcium, and promoted somatostatin secretion, which regulated insulin secretion from β cells [70]. Adriaenssens’s group also clarified that ghrelin administration to mouse islets increased somatostatin release and decreased insulin and glucagon releases. The decreased insulin and glucagon releases did not occur by ghrelin administration under blockade of somatostatin receptor [71]. On the other hand, some studies have indicated that pancreatic AG might contribute to regulating β cell survival and function by affecting cell proliferation and growth and inhibiting apoptosis [72,73]. It is unclear if these AG functions occur via binding to GHS-R. Unlike AG, UAG is not involved in increasing blood glucose levels and decreasing plasma insulin levels, but coadministration of UAG and AG improves insulin sensitivity [74] (Figure 2). Delhanty and colleague also revealed that UAG improves insulin sensitivity by modulating lipid metabolism [75]. They considered that the modulation is brought by suppression of various insulin resistant-correlated genes, including *Lcn2* [76] and *Ucp2* [77]. On the other hand, some studies have denied the improvement of insulin sensitivity by UAG [78,79]. Furthermore, UAG increased islet cell mass, promoted the survival of β cells, and prevented the induction of diabetes mellitus in streptozotocin-treated rats [80,81]. OB released from the pancreas plays an important role in the development and homeostasis of the pancreas [62] and in the generation of β cells from pancreatic precursor cells in mice [82]. OB also stimulates insulin release from β cell lines, inhibits apoptosis of β cells, and contributes to β-cell survival [83] (Figure 2). The prevention of β-cell apoptosis by OB administration was proven in a type 2 diabetes animal model [84]. Moreover, a recent study revealed that OB had the potential to regenerate insulin-producing cells from mesenchymal stem cells [85].

In summary, ε cells influence other endocrine cells, especially β cells, by releasing ghrelin. Ghrelin suppresses insulin release from β cells and increases blood glucose levels as well as contributes to cellular growth and inhibits apoptosis of β cells. These complex functions of ghrelin might be due to the existence of different subtypes, and further studies are needed to elucidate its mechanisms.

## 5. Conclusions

Almost 20 years have passed since the existence of pancreatic ε cells was confirmed, and many aspects of their characteristics, mechanisms of differentiation, and regulatory roles in other cells have been clarified. Importantly, ε cells have been shown to regulate various functions of β cells, including the control of blood glucose levels and cellular growth. These results suggest that strategies targeting ε cells may provide new treatment options for patients with diabetes mellitus. For example, regulation of acylated ghrelin released from epsilon cells may contribute to the improvement of insulin release and stabilization of blood glucose levels.

## Figures and Tables

**Figure 1 ijms-20-01867-f001:**
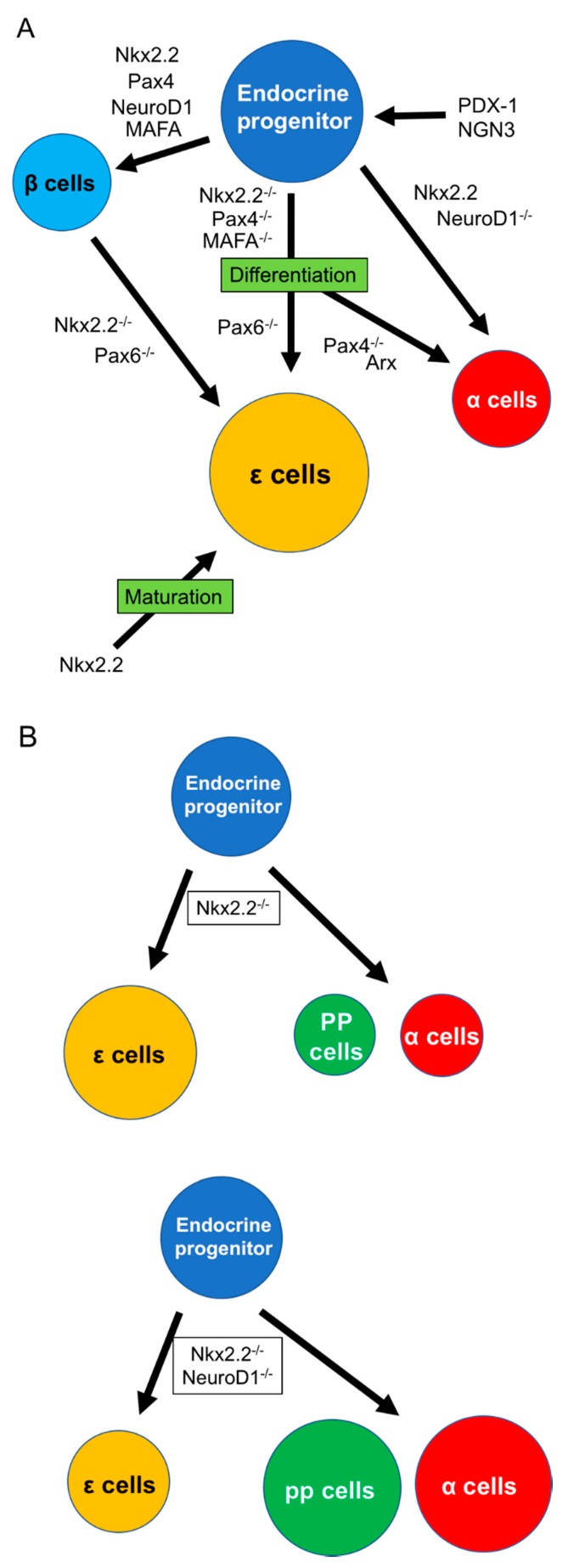
Differentiation of pancreatic ε cells. (**A**) Differentiation into various endocrine cells, including ε cells from endocrine progenitors, which receive the stimulation of PDX-1 and NGN-3. Differentiation of ε cells needs the inhibition of various transcription factors, including Nkx2.2, Pax4, Pax6, or MAFA. (**B**) NeuroD1 contributes to shifting the population between ε and α/PP cells under the Nkx2.2 null condition. Arx: aristaless-related homeobox, MAFA: V-maf musculoaponeurotic fibrosarcoma oncogene homolog A, NGN3: neurogenin 3, Nkx2.2: Nk2 homeobox, Pax4: paired box protein Pax-4, Pax6: paired box protein Pax-6, PDX-1: pancreas/duodenum homeobox protein 1, PP: pancreatic polypeptide.

**Figure 2 ijms-20-01867-f002:**
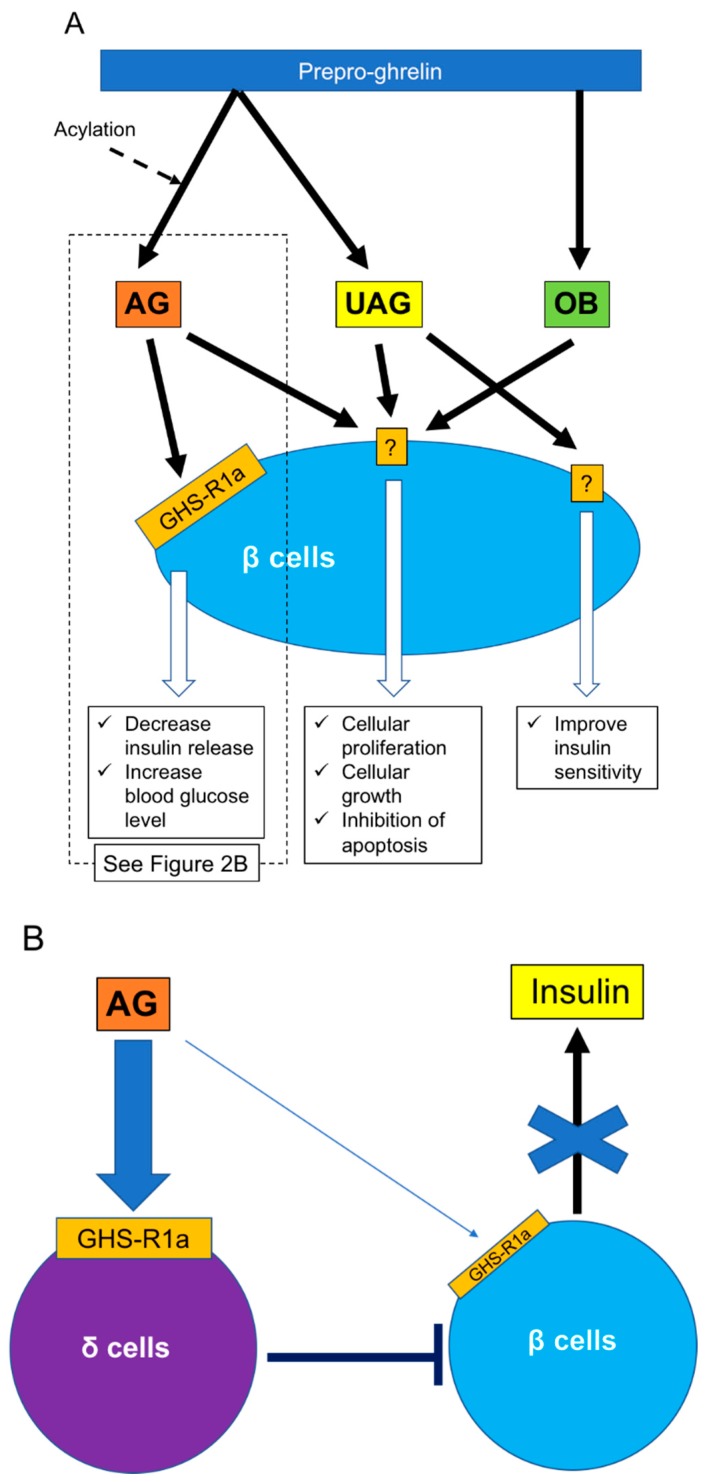
Effects of ghrelin for β cells. (**A**) Three ghrelin subtypes (AG, UAG, OB) regulate β cell functions, including insulin release, cellular proliferation and growth, and insulin sensitivity. (**B**) Regarding insulin release, it is clear that insulin release receives regulation from δ cells under AG stimulation rather than direct stimulation of AG to β cells. AG: acylated ghrelin, GHS-R: growth hormone secretagog receptor, OB: obestatin, UAG: unacylated ghrelin.

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
