# Peer review of "Development and Characteristics of Pancreatic Epsilon Cells"

_ijms, 2019, doi:10.3390/ijms20081867_

Round 1

Reviewer 1 Report

The authors have appropriately addressed all concerns raised by this reviewer, rendering the manuscript much more comprehensive and thorough.

Nonetheless, the new paragraphs included in the revision, clearly need to be edited for proper English.

Author Response

Please see attached comments.

Reviewer 2 Report

The manuscript is now acceptable.

Author Response

Please see attached comments.

This manuscript is a resubmission of an earlier submission. The following is a list of the peer review reports and author responses from that submission.

Round 1

Reviewer 1 Report

This brief review summarizes the development and characteristics of pancreatic epsilon cells initially identified in islets in the early 2000s. Although of interest, the manuscript falls short in its goal to summarize the biology/physiology of epsilon cells and their implication in diabetes.

1)    Section 3: A more in depth and comprehensive description of epsilon cell fate determination during pancreatic islet development accompanied by a relevant figure containing all pertinent factors is mandatory. For example, NEUROD and MAFA are missing from the current figure. Of not, this figure provides very little insight in its current format. Furthermore, the reader is completely confused on whether NKX2.2 expression or repression is required for epsilon cell fate. Finally, the authors own opinions would be welcome. For example, they could provide some clues on why epsilon cells comprise up to 30% of islet cells during development but then falls to almost nil in adult islets.

2)    Section 4: The authors should provide a better insight on the interaction of epsilon cells with all other islet endocrine cells and not only beta cells. To reinforce this point, the GHS-R1a is predominantly expressed on alpha cells. Thus the main target of acylated Ghrelin should be alpha and not beta cells. In this context, Figure 2 needs to be redone, to include all endocrine cells. A few curiosities: What dictates acylation of Ghrelin? What controls ghrelin versus OB biosynthesis? How does unacylated Ghrelin through beta cells improve insulin sensitivity? How do the authors envisage a potential diabetes treatment targeting the epsilon cells?

Author Response

This brief review summarizes the development and characteristics of pancreatic epsilon cells initially identified in islets in the early 2000s. Although of interest, the manuscript falls short in its goal to summarize the biology/physiology of epsilon cells and their implication in diabetes.

-       As you say, this review focused on development and characteristics of pancreatic epsilon cells, especially in their development, biology, physiology and possibility for treating diabetes. While many researches about pancreatic beta cells are proceeded, characteristics of other four endocrine cells are not fully examined, especially in pancreatic epsilon cells. The characteristics of epsilon cells are not fully understood even if specialists in pancreas. Thus, we aim to summarize the recent findings about epsilon cells including the development, interaction to other endocrine cells and possibility for treating diabetes. Certainly, the sources of information about epsilon cells are based on the studies in early 2000s and might be biased. In the revised version, novel findings in recent studies about epsilon cells are inserted.

1)            Section 3: A more in depth and comprehensive description of epsilon cell fate determination during pancreatic islet development accompanied by a relevant figure containing all pertinent factors is mandatory. For example, NEUROD and MAFA are missing from the current figure. Of not, this figure provides very little insight in its current format.

-       We insert a recent study about interaction between Nkx2.2 and NeuoD1 in epsilon cells development in revised version (1) (Page 5 Line 171-180). A study about development of ε cells under absent MAFA condition is also included (2) (Page 3 Line 126-135). Figure 1 is remade and new Figure 1B is included following the revision. Comments about Pax6 (3), which is shown in minimum in the original version, is also included (Page 5 Line 195-Page 6 Line 203).

Furthermore, the reader is completely confused on whether NKX2.2 expression or repression is required for epsilon cell fate.

-       The roles of epsilon cells are summarized in Section 3 for prevention of the confusion.

Finally, the authors own opinions would be welcome. For example, they could provide some clues on why epsilon cells comprise up to 30% of islet cells during development but then falls to almost nil in adult islets.

-       We consider the increase and decrease of epsilon cells in gestation are influenced by the expression patterns of PDX-1, NGN-3 and MAFA. Our idea about the change of epsilon cell population is inserted in Section 3 (Page 3 Line 129-135).

2)            Section 4: The authors should provide a better insight on the interaction of epsilon cells with all other islet endocrine cells and not only beta cells. To reinforce this point, the GHS-R1a is predominantly expressed on alpha cells. Thus the main target of acylated Ghrelin should be alpha and not beta cells. In this context, Figure 2 needs to be redone, to include all endocrine cells.

-       Recent study revealed that GHS-R on delta cells mainly contributed to the decrease of insulin release, rather than GHS-R on beta cells. We inserted the explanations and new Figure 2 in the revised version.

A few curiosities: What dictates acylation of Ghrelin? What controls ghrelin versus OB biosynthesis? How does unacylated Ghrelin through beta cells improve insulin sensitivity? How do the authors envisage a potential diabetes treatment targeting the epsilon cells?

-       Acylation of ghrelin is regulated by ghrelin O- acyl-transferase (Page 6 Line 232). According to our investigations, a factor which cleaves OB from prepro-ghrelin is unclear (Page 6 Line 237). Delhanty and colleague also revealed that unacylated ghrelin improves insulin sensitivity by modulating lipid metabolism (4). They considered that the modulation is brought by suppression of various insulin resistant-correlated genes including Lcn2 (5) and Ucp2 (6). On the other hand, some studies denied the improvement of insulin sensitivity by unacylated ghrelin (7, 8) (Page 8 Line 288-292). We consider that regulation of acylated ghrelin released from epsilon cells may contribute to improvement of insulin release and stabilization of blood glucose level (Page 8 Line 313-314).

-        

1.           Mastracci TL, Anderson KR, Papizan JB, Sussel L. Regulation of Neurod1 contributes to the lineage potential of Neurogenin3+ endocrine precursor cells in the pancreas. PLoS Genet. 2013;9(2):e1003278.

2.           Bruin JE, Saber N, O'Dwyer S, Fox JK, Mojibian M, Arora P, et al. Hypothyroidism Impairs Human Stem Cell-Derived Pancreatic Progenitor Cell Maturation in Mice. Diabetes. 2016;65(5):1297-309.

3.           Swisa A, Avrahami D, Eden N, Zhang J, Feleke E, Dahan T, et al. PAX6 maintains beta cell identity by repressing genes of alternative islet cell types. J Clin Invest. 2017;127(1):230-43.

4.           Delhanty PJ, Sun Y, Visser JA, van Kerkwijk A, Huisman M, van Ijcken WF, et al. Unacylated ghrelin rapidly modulates lipogenic and insulin signaling pathway gene expression in metabolically active tissues of GHSR deleted mice. PLoS One. 2010;5(7):e11749.

5.           Yan QW, Yang Q, Mody N, Graham TE, Hsu CH, Xu Z, et al. The adipokine lipocalin 2 is regulated by obesity and promotes insulin resistance. Diabetes. 2007;56(10):2533-40.

6.           De Souza CT, Araujo EP, Stoppiglia LF, Pauli JR, Ropelle E, Rocco SA, et al. Inhibition of UCP2 expression reverses diet-induced diabetes mellitus by effects on both insulin secretion and action. FASEB J. 2007;21(4):1153-63.

7.           Vestergaard ET, Jessen N, Moller N, Jorgensen JOL. Unacylated Ghrelin does not Acutely Affect Substrate Metabolism or Insulin Sensitivity in Men with Type 2 Diabetes. J Clin Endocrinol Metab. 2019.

8.           Tong J, Davis HW, Summer S, Benoit SC, Haque A, Bidlingmaier M, et al. Acute administration of unacylated ghrelin has no effect on Basal or stimulated insulin secretion in healthy humans. Diabetes. 2014;63(7):2309-19.

Reviewer 2 Report

This is a short review regarding development and characteristics of pancreatic epsilon cells. The manuscript was concisely and well written. Easy to follow.

It will be better to add short explanation for figure to the figure legends of Figures 1 and 2.

Author Response

This is a short review regarding development and characteristics of pancreatic epsilon cells. The manuscript was concisely and well written. Easy to follow.

It will be better to add short explanation for figure to the figure legends of Figures 1 and 2.

- Thank you. Short explanations for the figures are inserted in the revised version.